# MsWH: A Multi-Sensory Hardware Platform for Capturing and Analyzing Physiological Emotional Signals

**DOI:** 10.3390/s22155775

**Published:** 2022-08-02

**Authors:** David Asiain, Jesús Ponce de León, José Ramón Beltrán

**Affiliations:** 1Department of Electronics, Escuela Universitaria Politécnica de La Almunia, La Almunia de Doña Godina, 50100 Zaragoza, Spain; dasiain@unizar.es; 2Department of Electronics Engineering and Communications, Escuela de Ingeniería y Arquitectura, I3A, Universidad de Zaragoza, 50018 Zaragoza, Spain; jrbelbla@unizar.es

**Keywords:** multi-sensory platform, sensoring, skin temperature, blood oxygen saturation, heart rate, attitude and heading reference system, inertial measuring unit, electrodermal activity, bioimpedance, wearable devices, emotions

## Abstract

This paper presents a new physiological signal acquisition multi-sensory platform for emotion detection: Multi-sensor Wearable Headband (MsWH). The system is capable of recording and analyzing five different physiological signals: skin temperature, blood oxygen saturation, heart rate (and its variation), movement/position of the user (more specifically of his/her head) and electrodermal activity/bioimpedance. The measurement system is complemented by a porthole camera positioned in such a way that the viewing area remains constant. Thus, the user’s face will remain centered regardless of its position and movement, increasing the accuracy of facial expression recognition algorithms. This work specifies the technical characteristics of the developed device, paying special attention to both the hardware used (sensors, conditioning, microprocessors, connections) and the software, which is optimized for accurate and massive data acquisition. Although the information can be partially processed inside the device itself, the system is capable of sending information via Wi-Fi, with a very high data transfer rate, in case external processing is required. The most important features of the developed platform have been compared with those of a proven wearable device, namely the Empatica E4 wristband, in those measurements in which this is possible.

## 1. Introduction

Physiological signal capturing devices have evolved rapidly in recent years. From being a curiosity a decade ago, today, almost any electronic devices (mainly cell phones, smart watches or wearable devices in general) have several electronic sensors capable of recording the individual’s physical activity and registering physiological signals such as body temperature or heart rate. The capture of physiological signals itself has been the subject of extensive study in recent years [1], and the variety and importance of its applications has grown steadily throughout this time, especially after the COVID-19 pandemic [2].

The advent of all kind of non-invasive sensors has enabled the development of many platforms for acquiring an increasingly wide range of physiological signals. Examples of commonly measured physiological signals are electrocardiogram (ECG) [3,4,5,6,7], electromyogram (EMG) [8,9,10,11], photoplethysmography (PPG) [12,13], including heart rate (HR) and its variability (HRV) [14,15], body or skin temperature (ST) [16,17], skin conductance and impedance [18,19,20], electrodermal activity (EDA) [21,22,23], oxygen saturation (SpO2) [24,25,26], blood pressure (BP) [27], and respiration rate (RR) [28], among others.

### 1.1. Signal Capturing and Wearables

Wearable devices are intelligent electronic platforms, incorporated into clothing or worn on the body, generally in the form of accessories (forehead bands, bracelets, patches, watches), which allow the collection of physiological data from the user and, in general, a more or less detailed analysis of these data, with fewer restrictions compared to traditional laboratory devices [29]. Today, the use of wearable devices has become widespread, as have the signals that these devices are capable of recording and analyzing.

The first wearable device could be the pocket watch (16th century), which evolved into the wristwatch (1904) and the first digital watches (1969). Today, in the space of a wristwatch, almost all the capabilities of a cell phone can be integrated, including the ability to send and receive calls and messages and the use of apps, as well as different types of sensors that record everything from the subject’s physical activity to some of their basic physiological signals. In addition to the wristwatch [30], other wearable devices have taken the form of rings [17], earrings [31], forehead bands [32] and different clothing such as gloves, shoes and others [33].

Some examples of successful wearable devices that are capable of storing, analyzing and sending physiological signals are the Samsung Gear S3 Frontier watch (https://www.samsung.com/es/wearables/gear-s3/highlights/ Accessed on 1 August 2022) (capable of measuring position and motion and HR, and featuring barometer, GPS and light sensor), the MUSE and MUSE 2 forehead bands (https://choosemuse.com/ Accessed on 1 August 2022) (which record and analyze EEG signal, as well as being able to measure HR, SpO2, motion and position) and the Oura rings (https://ouraring.com/ Accessed on 1 August 2022) (with heart rate monitoring, activity detection and seven temperature sensors). For reasons that will become evident later, it is necessary to mention one of the most successful wearable devices on the market today, the Empatica E4 wristband (https://www.empatica.com/en-eu/research/e4/ Accessed on 1 August 2022), which has four sensors: PPG sensor, EDA sensor, a three-axis accelerometer and an optical thermometer for ST measurement. This device has several working modes including data recording and transfer, streaming mode for real-time data visualization and E4 connect mode, which visualizes and manages physiological data (encrypted) in a cell phone app.

The development of platforms for recording, monitoring and analyzing physiological signals has continued growing for more than a decade [1,34] and is now a very active branch of sensor electronics [35,36,37,38]. The range of uses of captured physiological signals has expanded proportionally to the spread of this technology, from the extraction of health-related characteristics to the classification of emotions or moods. More specific applications of physiological signals may include, among others, disease diagnosis [25,39], remote healthcare [40,41,42,43,44,45,46], human–machine interaction (e.g., to increase the autonomy of people with disabilities) [47,48], rehabilitation processes [49], stress detection [34,50,51], monitoring of daily activity [52,53] and emotion detection [32,38,54,55,56,57].

This paper will detail the development of a platform for the acquisition of physiological signals, which includes a total of five different biomedical signals: skin temperature (ST), blood oxygen saturation (SpO2), heart rate monitoring (HR), movement/position of user’s head (AHRS) and electrodermal activity (EDA). The sensors used for the different measurements have been chosen among the most advanced ones available on the market, trying to prioritize the accuracy of the measurements as well as their performance. For example, the EDA sensor can also measure bioimpedance and allows different types of measurement configuration, including AC and DC (unipolar and switched bipolar), which improves the signal-to-noise ratio. As will be explained later, the accuracy of the system has been improved by a measurement based on two-channel multiplexing. A complete AHRS system has been implemented, including a three-axis accelerometer, gyroscope and magnetometer. The sensor for measuring SpO2 and HR is also state-of-the-art, allowing high-precision measurements. In addition, a fisheye camera is incorporated for the study of facial expression. Although the present work focuses on the electronic part of the design, the main objective is to use the Multi-sensory Wearable Headband (MsWH) platform for emotion recognition. Indeed, in emotion recognition, valence and arousal (among other emotional parameters) can be obtained through the selected physiological signals [58,59] and facial expression [55,60,61,62]. However, this device also has the potential to be used to detect and discriminate acute psychological stress and the performance of different physical activities occurring at different times or concurrently [50,63].

### 1.2. Aims of This Paper

As mentioned above, this paper will detail the development of the Multi-sensory Wearable Headband (MsWH) platform, which is capable of recording and analyzing five different physiological signals (ST, SpO2, HR, IMU and EDA), in addition to capturing the user’s facial expression using a fisheye camera. The MsHW platform integrates sensors already calibrated by the manufacturers (ST, HR, SpO2 and IMU). For this reason, aspects such as device consumption, EDA signal reading and response time of the temperature sensors are dealt with in greater depth in this work. In these measurements, the tests have been performed with accurate measurement systems on test bench. On the other hand, as a validation of results, the available compatible signals such as EDA, ST and HR have been compared against the Empatica E4 wristband data.

In the following section (Section 1.1), the technological differences of the developed platform compared to similar devices will be detailed. The present work focuses basically on the electronics required to capture the different signals, first treated in a descriptive way (Section 1.1). The necessary power supplies, the conditioning stages and the connection to and between the different integrated circuits capable of recording and analyzing (totally or partially) the raw signals are described and detailed in Section 3. In Section 4, there is a brief description of the software required to receive and interpret these signals. The results obtained in the capture of these signals are shown in Section 5, in which the data obtained by the device presented are compared with those obtained through the Empatica E4 wristband when possible. Finally, some conclusions and the future work are presented in Section 6.    

### 1.3. Related Works

This section will briefly detail some works on devices similar in one or more aspects to the one detailed here in order to highlight the differences of the proposed platform with respect to other pre-existing ones.

There is a wide variability of integrated circuits capable of performing accurate ST measurement. In [32], the authors use the LMT70 integrated contact temperature sensor, which is suitable for portable devices and has an error as low as 0.05 °C. In [23,37], the MLX90614 chip is used. This integrated sensor uses infrared technology, presents an error in the ST measurement of 0.5 °C, communicates by SMBus and has a library for use with Arduino. As will be seen in Section 3.1, the MAX30205 has been used in the MsHW platform, with an accuracy of 0.1 °C in the range of 37 to 39 °C and sends data via an I2C-compatible interface.

As will be seen in Section 3.2, for HR and SpO2 measurement, we have used the MAX30101 integrated unit, which is a similar but more modern version than, for example, the MAX30100 integrated unit used in [23].

The most significant advantage in terms of user motion measurement in the present work (Section 3.3) is that MsHW is provided with a complete AHRS system. Indeed, the BNO055 has a three-axis accelerometer, gyroscope and magnetometer, unlike other [55] platforms, where the latter does not exist. Thus, a complete inertial system is available that is capable of recording relative position and movements with high accuracy.

The EDA measure is currently one of the most widespread and active in research. Most devices present a solution for EDA measurement based on conventional analog circuitry (in the sense of circuitry built basically from amplifiers). For example, in [31] a three-level amplifier stage (3CA) has been designed: a first unity gain stage with an AD8244, a preamplifier stage with an AD8222 and a 24-bit low-noise ADC (ADS1299), plus a final adaptive amplifier stage with an MSP432. In [32], a direct impedance measurement is performed with a circuit where 2 low-power MCP6422 amplifiers are used. In [22], a single AD8603 is used, while a solution more similar to the one presented here is given in [37], with a double impedance measurement system through two MPC6004, plus a second stage that adjusts the output voltage to that of the ADC range used for signal processing.

In this work, a more compact solution has been chosen. The AD5941 is a measurement system that can be used, among other applications, for electrochemical measurements, battery impedance, glucose monitoring and bioimpedance measurements. It includes the analog inputs for the corresponding sensors, the conditioning of the input signals (amplifying, filtering), the generator of the excitation signals, and the converters and the microcontroller required for the calculation of the variable to be measured.

In conclusion, in the MsHW platform, we have tried to use state-of-the-art technologies in the precision measurement of the corresponding signals, with the idea of having highly accurate data that will be used later for emotion recognition. In this sense, it is known [64] that multimodal emotional detection systems tend to be more accurate than unimodal ones. As concluded in [65], the combination of methods for recording and analyzing physiological variables (especially if it is a combination of several signals) with deep learning algorithms can be an extremely powerful tool in the detection of emotional states in any of its applications.

## 2. Platform Overview

As mentioned in the previous section, the present work will detail the development of a platform that includes a total of five different physiological signals: skin temperature (ST), blood oxygen saturation (SpO2), heart rate (HR) monitoring, user movement/position (IMU) and electrodermal activity/bioimpedance (EDA). In addition, a fisheye camera is incorporated for facial recognition and facial expression study. The device is attached to the subject’s head by means of an elastic band and to the earlobe by means of a clip.In this sense, the developed platform is “wearable”, but it has been developed for data collection in the laboratory or under controlled conditions, and it is not for everyday use in free living or physical activity. On the forehead, on the same bracket that holds the camera, are located the power and communications control system (Wi-Fi, Bluetooth, and a serial-USB port), the main microcontroller, the chip for EDA, the gyroscope and accelerometers (IMU), and the LIPO battery that powers all the electronics. The platform is completed with a second board (oximeter) that is placed in the earlobe. The information from all these sensors can be preprocessed inside the device itself (ST, EDA, IMU, HR, SpO2) or the raw data can be sent to a microcontroller capable of analyzing it and extracting high-level characteristics.

The complete system can be seen in Figure 1. The developed platform is divided into two main subsystems or PCBs. Figure 1a shows a side view of the platform, which is attached to the subject’s forehead by means of an elastic strap. The image also reflects the approximate view area of the camera. Figure 1b shows the electrodes of the EDA/bioimpedance subsystem. These electrodes (the silver brackets attached to the forehead and perfectly static by means of a strip of ethyl–vinyl acetate or EVA foam). In green, the temperature sensor can be seen. Figure 1c shows an identical view to the previous one with the electrodes placed in their final position and fixed by means of the EVA foam. This way, they will be in contact with the subject’s forehead and always separated at the same distance. The second PCB, attached to the ear by means of a clamp, is shown in Figure 1d. This PCB contains the SpO2 and HR measure systems, as well as the fastening system (in pink). In these images, we can be mainly see a (1) microcontroller, (2) camera, (3) LIPO battery, (4) EDA electrode, (5) pulse oximeter, (6) temperature sensor.

Each of these subsystems has its own power supply as well as the signal conditioning stages necessary for its correct operation. In addition, both modules have a microprocessor (ESP32) capable of analyzing in situ the signals from the different sensors and extracting the most notable characteristics from them, although if necessary, both devices can, in turn, dump the data in raw format to an external computer for further processing via Bluetooth and Wi-Fi connection.

The total weight of the platform is 168 grams, 96 grams of which correspond to the head attachment system. The PCBs of both subsystems can be seen in Figure 2. On the left side, Figure 2a,b show the top and bottom views of the PCB containing the RS232 system, the camera connection port (bottom view), the main microcontroller (wich includes Wi-Fi), the communication and battery control, the switched-mode power supplies that generate the supply voltages for the different sensors and their conditioning, and a USB port (on the right side of the board, top view). The size of this PCB is 38 mm × 124 mm. Figure 2b,d, on the right, show the top and bottom faces of the PCB corresponding to the pulse oximeter, which is attached to the subject’s earlobe by means of a clamp, as shown in Figure 1d. The size of this PCB is 18 mm × 20 mm. There is a third very simple PCB, which contains only the ST sensor and must be separated from the others since the sensor used needs to be in thermal contact with the subject’s forehead. This small PCB can be partially seen in Figure 1b,c, number (6).

The following section will detail the measurement devices for each of the physiological signals implemented in the platform that has been developed.    

## 3. Hardware Description

In this section, we detail the proposed platform design in terms of hardware and signal acquisition. The block diagram of the platform is shown in Figure 3.

As can be seen in this figure, the platform is powered by lithium–polymer batteries with output voltages of 5.0 V and 3.3 V for the different devices. The EDA sensor, the IMU and digital camera are located inside the forehead board. The temperature sensor and pulse oximeter are located in the earlobe subcircuit. These two devices and the IMU communicate with the main microcontroller via I2C, while the EDA sensor sends its measurements via SPI and the camera uses the parallel port. The main microcontroller can, in turn, communicate with external devices via Wi-Fi or Bluetooth. The approximate battery life, with all sensors and camera at full power, is about 2 h. In this time, the amount of data collected, analyzed and sent can be massive.

Some characteristics of the different sensors and devices included in the platform are shown in Table 1. Each subsystem will be detailed in the following sections.

### 3.1. Skin Temperature (ST)

Attached to the earlobe, the device contains the MAX30205 (https://datasheets.maximintegrated.com/en/ds/MAX30205.pdf, accessed on 14 June 2022) skin temperature sensor, which, as advanced before, has a 16-bit (0.00390625 °C) temperature resolution, and a final clinical grade accuracy of 0.1 °C from 37 to 39 °C. This device converts analog temperature measurements to the corresponding digital data via a high-resolution sigma-delta analog-to-digital converter (ADC). The measurements obtained can be sent via a lockup-protected I2C-compatible two-wire serial interface. The sensor has a supply voltage range of 2.7 to 3.3 V and a low supply current of 600 µA.

The main difficulty in measuring temperature using a wearable device is making sure that the sensor is measuring the body temperature and not the temperature of the device itself, or whether the measurement is excessively influenced by the ambient temperature. These objectives can be achieved by maximizing skin contact, minimizing the thermal mass of the circuit (including the PCB), and minimizing the thermal exchange between the device and its enclosure with the environment. Figure 4 depicts the detailed circuit diagram of the MAX30205.

### 3.2. Blood Oxygen Saturation (SpO2) and Heart Rate (HR) Monitoring

The block diagram of the pulse oximeter and heart rate monitoring are shown in Figure 5. The monitored data are sent to a MAX32664 (https://datasheets.maximintegrated.com/en/ds/MAX32664.pdf, accessed on 14 June 2022) micro-controller. It is a sensor hub family with integrated firmware and world-class algorithms for wearables. The MAX32664 seamlessly enables any desired sensor functionality and includes communication with Maxim’s optical sensors. It can deliver raw or already processed data to the outside world through a fast-mode slave I2C interface. A firmware bootloader is also provided.

The MAX32664 firmware version for the developed platform is the so-called Version C. It supports the MAX30101 high-sensitivity pulse oximeter and heart-rate sensor for wearable health. It also supports estimated blood pressure monitoring. The MAX30101 is an integrated module that includes internal LEDs (red and IR for sensing and green for ambient light cancellation), photodetectors, and other optical elements, with low-noise electronics. The MAX30101 provides a complete system solution to ease the design-in process for mobile and portable devices. As shown in Figure 5, it requires a 1.8 V power supply (provided by a SPX3809), plus a separate 5.0 V power supply for the internal LEDs (obtained using a Buck-Boost REG71050 DC-DC regulator). As previously advanced, communication with the MAX32664 is performed through a standard I2C interface. The module can be shut down by software with zero standby current, which allows energy saving.

The complete circuit diagram of the pulse oximeter is shown in Figure 6. The connections between the different integrated circuits, as well as other external elements necessary for their correct operation, can be observed.

### 3.3. Attitude and Heading Reference System (AHRS)

The subject’s orientation, position and movement (Attitude and Heading Reference System, AHRS) are monitored by the Inertial Measuring Unit (IMU) BNO055 (https://www.bosch-sensortec.com/media/boschsensortec/downloads/datasheets/bst-bno055-ds000.pdf, accessed on 14 June 2022), which is an absolute orientation sensor. It is a System in Package (SiP), integrating a 14-bit three-axis accelerometer, a 16-bit three-axis gyroscope (with a range of ±2000 degrees per second), and a three-axis magnetometer. These three sensors communicate via SPI with a 32-bit Cortex M0+ microcontroller running Bosch Sensortec sensor fusion software. The BNO055 includes internal algorithms to constantly calibrate the gyroscope, accelerometer and magnetometer inside the device. The BNO055 is equipped with bidirectional I2C and UART interfaces. Once programmed to run with the HID-I2C protocol, it turns into a plug-and-play sensor hub solution. The corresponding circuit diagram is shown in Figure 7.

The final image size is 640 × 480 pixels. In addition, the OV2640 module performs on-chip JPEG encoding, converting the pixels into a machine-readable format without additional computing from the MCU. The camera works at 12.5 frames per second.

### 3.4. Face and Gesture Recognition

To perform gesture and face recognition, which is a very important characteristic in emotion detection and classification, the OV2640 (https://www.uctronics.com/download/cam_module/OV2640DS.pdf, accessed on 14 June 2022) camera module was chosen for its high sampling rate and variable pixel resolution. As mentioned above, the camera is fixed on a very light plastic support, which protrudes from the subject’s forehead (see Figure 1). Thus, the viewing area remains approximately constant, regardless of the user’s movements, which is expected to increase the accuracy of the results of the facial expression recognition algorithms.

### 3.5. Electro Dermal Activity (EDA) and Bioimpedance

Human skin puts up a certain resistance to the flow of electric current, and therefore, it has an associated electric impedance that can be experimentally measured. The bioimpedance depends on multiple variables both prior to the measurement (such as the separation between the electrodes) and dependent on external stimuli (such as the dermal response to variations in body temperature or perspiration).

As advanced in Section 1.1, the Electro-Dermal Activity is obtained through the integrated circuit AD5941 (https://www.analog.com/media/en/technical-documentation/data-sheets/ad5940-5941.pdf, Acessed on 1 August 2022). This integrated contains the complete EDA/bioimpedance measurement system that will be detailed below.

#### 3.5.1. Electrodes Selection and Configuration

The bioimpedance measurement process is carried out by means of electrodes (from two to four) that can be placed on different parts of the body, separated by a certain distance. In turn, electrodes can be divided into two broad categories: dry and wet. The measurement procedure is different in each case. To make it user-friendly, dry electrodes have been chosen for the developed platform, while the electrical excitation of the skin is carried out by means of a low-voltage sinusoidal signal at 100 Hz frequency, which the integrated circuit itself is capable of generating. As can be seen in Figure 1, the electrodes are placed on the forehead, close to the measuring device itself, located in the main PCB. This location of the electrodes has two advantages: firstly, the forehead is one of the areas of greatest electrodermal activity [66], making measurements in this area particularly sensitive and/or accurate. Secondly, the electrodes remain in a more stable position with respect to, for example, the wrist, thus reducing the risk of electrical noise and unwanted signal variations, which cloud the final quality of the captured data.

In any case, the bioimpedance measurement must meet the IEC 60601 standard, which limits the current that can flow through the human body to a maximum of 10 µA for frequencies below 1 kHz. According to this standard, the sensors must be isolated by a decoupling capacitor $C_{ISOx}$ (where *x* is the wire number) which guarantees that no DC current penetrates the body. The input current is limited by a resistor $R_{LIMITx}$. The connection and the final result of the electrode impedance are shown in Figure 8.

As can be seen in Figure 8, there are three main different ways to measure skin bioimpedance, depending on the type of measurement required in each case: through a two-wire, three-wire or four-wire connection. The values of the different decoupling capacitors ($C_{26}$ to $C_{30}$) and limiting resistors ($R_{16}$ to $R_{19}$) for both connections can be seen in the figure. In all measurement cases, the bioimpedance is essentially located between the electrodes connected to the CEO and AIN1 terminals of the measuring device (see Figure 9, below), which are decoupled in Figure 8 by capacitors $C_{26}$ and $C_{30}$, respectively. In the four-wire measurement, the remaining electrodes are connected in the proximity of the previous ones, so that the detected impedance is essentially the same.

Although the two-wire and four-wire measurements mentioned above are the most commonly used, in our case, a three-wire connection has been chosen. In this case, a reference electrode is taken, while the skin impedance measurement is obtained between the reference electrode and one of the two additional ones, all of them placed on the individual’s forehead (see Figure 1). Given the short distance between the electrodes, two similar but slightly different impedances ($Z_{U_{1}}$ and $Z_{U_{2}}$) will be measured, whose value is multiplexed and analyzed in the microprocessor, seeking to take full advantage of the device’s capabilities (e.g., improving the data accuracy and its signal-to-noise ratio). The measurement process will be detailed in the next section.

#### 3.5.2. Bioimpedance Measurement Process

As mentioned above, the bioimpedance measurement is carried out by the AD5941 integrated circuit. The AD5941 is a high-precision, low-power analog front end, which is specifically designed for portable applications that require high-precision, electrochemical-based measurement techniques, such as current, voltage or impedance measurements. It has been designed for skin and body impedance measurements, and it works as a complete bioelectric or biopotential measurement system.

EDA is measured by voltammetry. To measure an unknown impedance, an excitation signal $V_{EX}$ is applied across this impedance. First, the voltage across the terminals of the unknown impedance is measured. In a second stage, the current flow through the unknown impedance is measured. This current is converted to a voltage via a Trans-Impedance Amplifier (TIA), and this voltage is measured by an Analog to Digital Converter (ADC). In a first stage of analysis, a Discrete Fourier Transform (DFT) is performed on the data measured by the ADC for the current and voltage values, thus obtaining the real and imaginary parts of $V_{Z_{U_{1,2}}}$ and $I_{Z_{U_{1,2}}}$. Although there are other methods to perform this calculus, DFT has been chosen since it is implemented in the AD5941 itself, and therefore, no additional operation or programming is required.

As advanced in the previous section, in our case, the bioimpedance measurement is carried out by means of a three-wire connection detailed in Figure 9. In this case, the electrode connected to input CEO (ELEC.♯1) is used as the reference electrode, while a selector (taken from the matrix of selectors of the integrated unit) selects the other point of the skin among the two remaining electrodes (ELEC.♯2 and ELEC.♯3 in the figure, which are connected to inputs AIN1 and AIN3, respectively) to obtain two different impedance measurements using the procedure described above.

Under these circumstances, the module of both measured impedances can be obtained through the expression:(1)$$|Z_{U_{1,2}}|=\frac{|V_{Z_{U_{1,2}}}|}{|I_{Z_{U_{1,2}}}|}=\frac{|V_{EX}|}{|V_{O_{TIA}}|}R_{TIA}$$

The excitation signal, $V_{EX}$, is a sinusoidal wave of 1.1 V amplitude, oscillating at a frequency of 100 Hz. The use of an AC signal improves the overall characteristics of the recorded signal compared to that obtained with a DC or a switched DC measurement. Knowing the characteristics of the excitation signal, it is possible to calculate the isolation capacitors necessary for the correct connection of the electrodes (Figure 8). For this purpose, a limiting resistor per channel is taken as follows:(2)$$R_{LIMIT}=1\;k\Omega$$

In order to match the body impedance analysis measurement, the return path capacitor $C_{ISO2}$ for EDA must have a value of:(3)$$C_{ISO2}=470\;\mathrm{nF}$$

With these values for $R_{LIMIT}$ and $C_{ISO2}$, the isolation capacitor $C_{ISO1}$ is calculated by means of the expression:(4)$$X_{C_{1}}<\sqrt{\left(\frac{V_{EXC_{RMS}}^{2}}{I_{AC_{RMSLIMIT}}^{2}}-R_{LIMIT}^{2}\right)}-X_{C_{2}}\Rightarrow C_{ISO1}=15\;\mathrm{nF}$$
where:$X_{C_{i}}=1/2\pi fC_{i}$ is the module of the impedance of capacitors $C_{ISO1}$ and $C_{ISO2}$.$V_{EXC_{RMS}}=0.778$ is the RMS value of a sinusoidal signal of 1.1 V amplitude.$I_{AC_{RMSLIMIT}}$ is the maximum allowed AC current in the human body (IEC 60601).$f_{EXC}=100\;\mathrm{Hz}$ is the frequency of the excitation signal.

The complete EDA measurement process can be seen in Figure 10. As advanced, the excitation signal is a sinusoid of 1.1 V amplitude and 100 Hz frequency. As can be seen in the figure, the complete process takes 250 ms. During the first approximately 100 ms, with the system clock working at 16 MHz, the multiplexed reading of the two channels takes place. The rest of the time the system remains hibernated. Specifically, during the first 50 ms (five full signal periods), the excitation and impedance reading is carried out through the left channel (blue in the figure). Then, by changing the switch mode in the switch matrix of the device (see Figure 9), the excitation and reading of the right channel (in orange in Figure 10) is performed. Then, the device hibernates until the next lecture cycle, about 150 ms later. During the hibernation period, the clock works at 32 MHz. The collected data (which include the FFT of the multiplexed read signals) are stored in a FIFO memory, which has a certain size (a user-defined threshold). When the FIFO buffer is full, the system sends a warning signal to the microprocessor (FIFO interrupt). Before the next read cycle, the MCU must find the time to receive the data, which is an event it will mark with a new interrupt. In this way, the FIFO buffer remains empty for a new storage cycle.

Finally, the detailed circuit of connections and external elements connected to the AD5941 can be seen in Figure 11.

One of the main advantages of this subsystem is the possibility of measuring both EDA and bioimpedance. On the other hand, the multiplexing of the measured impedances, $Z_{U_{1}}$ and $Z_{U_{2}}$, leads to an increase in the performance of the device.

## 4. Software Description

The software has been developed in the official framework for ESP32, the Espressif IoT Development Framework ESP-IDF (https://docs.espressif.com/projects/esp-idf/en/latest/esp32/, accessed on 14 June 2022). It has been programmed using both the Application Programming Interfaces (API’s) and the FreeRTOS operating system (https://www.freertos.org/, accessed on 14 June 2022), which are included in the same framework.

The design of the application can be seen in Figure 12. It works on a producer–consumer basis, connected by queues that act as temporary buffers for discontinuities, caused in the Wi-Fi interface due to connection quality or performance changes in the data transfer rate.

As can be seen in Figure 12, the thread producers are camera capture and signal acquisition. Both work asynchronously but with the same time reference (timestamp). The camera capture thread uses the ESP32-camera driver (https://github.com/espressif/esp32-camera, accessed on 14 June 2022). As advanced in Section 3.4, the capture rate is 12.5 frames per second in continuous mode for VGA resolution. For each frame, dynamic memory is reserved, and the pointer is queued. The thread consumer (video streaming in Figure 12) pulls the data out of the queue and sends them over the UDP socket to the target host. Finally, the memory is freed.

The second producer thread is the signal acquisition. This thread synchronizes and acquires signals from all the sensors of the platform. For this purpose, a synchronous acquisition start trigger is performed, and then, the transfers are attended with their different intervals based on the configuration of their internal buffers. In the case of the AD5941, the transfer is controlled by interruption every second with a buffer containing four samples. For the MAX30205 temperature, an acquisition is performed by polling every 250 ms. These two sensors are queued together in one structure, since they are acquired at the same sampling rate. In the same task, also by polling, the internal buffer of the SpO2 implemented in the MAX32664 processor is checked and the data are placed in a specific queue. The same operation is performed with the data obtained from the AHRS, BMO055. For each of the queues, a consumer thread is associated to send the data over UDP sockets on consecutive ports with respect to the base port to the IP address of the destination host.

The processor load is balanced between two cores. On Core 0, the video tasks are executed, and on Core 1, the acquisition threads are executed. The network threads are also configured in Core 1. This load balancing and the use of UDP sockets ensure a correct latency of the application. It should be noted that the application performance degrades if the TCP socket is used, not reaching 8 FPS compared to the stable performance of 12.5 FPS when using the UDP socket. In our UDP protocol, the timestamp and frame counter are included in the same way as in RTP (Real-time Transport Protocol), and the receiver must use them to detect packet loss and to accommodate out-of-service delivery.

In general, the platform is in the same local network as the server and the use of UDP does not require specific configuration in the routers.

Finally, there is a control command shell accessible via the TCP socket. These commands allow to start or stop the different data streams and to parameterize some additional configurations.

## 5. Experimental Results

In this section, the raw data from the different sensors and devices included in the developed platform are shown. Where possible, the equivalent results sampled by means of the Empatica E4 wristband are additionally shown. This comparison is of particular relevance for EDA, heart rate and body temperature. Although the Empatica has a three-axis accelerometer, it does not have a complete AHRS system. It is also unable to record facial expressions, so a comparison of results in the corresponding MsWH subsystems is not possible or relevant.

### 5.1. Power Consumption

This section provides the power consumption data for the different devices of the platform. The results are summarized in Table 2, in which the experimental measurements of power consumption are detailed in addition to their theoretical values, which are extracted from the data sheets.

### 5.2. Sensors and Calibration

As mentioned above, the ST (MAX30205), HR and SpO2 (MAX30101-MAX32664) and IMU (BNO055) sensors are factory calibrated, with measurement accuracy guaranteed by the manufacturer. The same is true for the OV2640 that controls the fisheye camera.

The facial recognition camera can work with 66° and 160° (fisheye) lenses. The distance between the camera lens and the individual’s forehead is 170 mm. Optical distortion can be corrected easily, since the distance to the plane remains constant throughout the recording. However, if necessary, a calibration standard can be used to more accurately correct for lens distortion. The frames acquired by the camera have been analyzed with the Amazon Rekognition Emotion API (https://docs.aws.amazon.com/rekognition/index.html, accessed on 14 June 2022), interpreting emotional expressions such as happiness, sadness and surprise with satisfactory results.

In the case of the ST and AHRS (IMU) measurements, the calibration process is included in the documentation detailed respectively in Section 3.1 and Section 3.3.

On the other hand, oxygen saturation measurement is calculated based on internal algorithms implemented in the MAX32664C sensor-hub, which is factory calibrated by default without optical shielding (https://www.maximintegrated.com/en/design/technical-documents/app-notes/6/6845.html, accessed on 14 June 2022) as used in the MsHW platform.

In the case of ST, HR and EDA, the signals obtained by the MsHW platform can be compared with those obtained by the Empatica E4 wristband. The results are shown in the following sections.

### 5.3. The Empatica E4 Wristband

The Empatica E4 wristband is a wearable wireless device designed for comfortable, continuous, real-time data acquisition in daily life. It contains four sensors:Photoplethysmography (PPG) to provide blood volume pulse, from which heart rate, heart rate variability, and other cardiovascular features may be derived.Electrodermal activity (EDA), used to measure sympathetic nervous system arousal and to derive features related to stress, engagement, and excitement.Three-axis accelerometer, to capture motion-based activity (IMU).Infrared thermometer, reading skin temperature (ST).

### 5.4. Data Capture Comparison

The comparison of the experimental data will be carried out in three different ways. Firstly, a comparison will be made in the measurement of skin temperature both from the point of view of the experimental results obtained and the response time (a relevant data, given the difference in the technologies used in both devices). Then, a comparison is made by obtaining the HR (or interval between heartbeats) in both devices. Finally, the results are obtained in EDA by measuring a set of resistances (or impedances) of known value, and the (more appropriate) direct measurement on the skin of an individual will be compared.

It should be noted that although both the developed platform and the Empatica E4 wristband have accelerometers, the results of the measurements with these sensors cannot be compared, since they are placed in different places on the body, and it is extremely difficult to design an experiment in which the movements of both devices are compatible and comparable.

#### 5.4.1. Skin Temperature

For skin temperature measurement, the resolution of the E4 is of 0.02 °C, while the MsWH resolution is of only 0.0039 °C. However, the accuracy of both systems turns out to be similar (±0.2 °C for the E4 and ±0.1 °C for the MsWH, both within 36–39 °C). The final difference in skin temperatures measured through E4 and MsWH differs by only 0.1 °C. This result was obtained by fixing the sensors of both devices against each other and leaving them at rest in the laboratory, thus measuring the ambient temperature.

Another comparative variable of interest is the Step Response Time (SRT), i.e., the time required to obtain an accurate measurement when faced with an abrupt variation of the variable to be measured. Using the Matlab System Identification Toolbox, and through the data obtained by both temperature sensors, the step response of each device has been generated. These responses are shown in Figure 13a,b.

As can be seen by comparing both subfigures, the SRTs of the compared devices are very different. This is due to the different technologies of the temperature sensors used in each case. In the E4 wristband, an optical thermometer is used. The absence of skin contact reduces the thermal inertia in the measurement, and therefore, the SRT is very low—less than 2 s, as can be seen in Figure 13a. On the other hand, as mentioned in Section 3.1, the MsWH uses a MAX30205, which requires direct contact with the skin and therefore has a higher thermal inertia. In this case, the SRT is almost 1 min, as can be seen in Figure 13b.

Although very significant, this difference in response times is not important because it is the measurement of skin temperature, and variations in skin temperature are not usually significant over such short time intervals for system thermal inertia to be an issue.

Figure 14 shows a more detailed comparison of the time response between the Empatica E4 wristband and our MsWH system in which the faster response of the Empatica E4 wristband can be clearly seen.

#### 5.4.2. Heart Rate

Previous studies [67] suggest that the Empatica E4 is potentially usable as a tool for investigating HR and its variability (HRV), at least while the user/patient is at rest. Nevertheless, the HR capture algorithms used in the case of the MAX32664 sensor hub and the Empatica wristband are very different. In the first case, it is an algorithm implemented within the device itself with a refresh rate of 100 Hz. In the case of the E4 wristband, the output data are provided by the device itself, which are calculated through the E4 streaming server software and sent through the Bluetooth connection. The Empatica captures the Inter-Beat Interval (IBI), eliminating unclear minimums due to the presence of noise. The remaining data are used to calculate the instantaneous heart rate and are sent via Bluetooth.

This makes signal comparison between these devices a complex process. The data supplied by Empatica must be filtered and subsampled, and the phase difference between the signals must be corrected before proceeding to a comparison of results.

The HR raw results are shown in Figure 15. The instantaneous heart rate of E4 (in red in the figure) has a higher variability than the data provided by MsHW (blue). This is due to the different calculation algorithms explained in the previous paragraph. Nevertheless, both measurements are comparable in the module, and variations are similar.

After processing the E4 wristband data (as explained before, subsampling, filtering, and synchronizing the E4 raw data), the HR measurement from both devices can be seen in Figure 16, which represents a longer time sequence than that in Figure 15. As can be seen in the figure, the measures of both systems are very similar, and the time delay is negligible. In this figure, the HR measurement of the MsWH is extracted directly from the raw data.

The difference between the MsWH and the E4 HR measurements can be calculated in percentage and plotted as a histogram, as has been shown in Figure 17.

As can be seen in this figure, the Gaussian fit shows that the difference in the HR measurement between these two devices is approximately of −0.14%, which can be used to validate the HR measurement of our system, assuming that the measurement of the E4 wristband is also accurate.

#### 5.4.3. Electro-Dermal Activity and Bioimpedance

The performance of the EDA and bioimpedance measurement systems has been verified by means of two simple experiments: in the first one, the correct calibration of the equipment is checked by measuring a set of seven precision resistors’ (250kΩ, 500kΩ, 1MΩ, 2MΩ, 3MΩ, 4MΩ and 5MΩ), with a 0.1% tolerance, which is equivalent to conductance values between 0.2μS and 4μS. In the second experiment, the measurements are carried out over the skin of the same subject.

The results of the calibration check of both devices are shown in Table 3. As mentioned above, we have a set of seven precision resistance values (or in other words, conductance values). These values are shown in the table (columns $R_{theo}$ and $S_{theo}$) and will be taken as merely informative. The complete calibration results are represented in Figure 18.

First, the resistors were measured with a high-performance multimeter (Fluke) to check their actual resistance and conductance values (columns $R_{F}$ and $S_{F}$ in Table 3). Next, the conductance values were measured using first the E4 wristband and then with our platform, MsWH. The results of these measurements are shown in the $S_{E4}$ and $S_{MsWH}$ columns. Based on the Fluke conductance results, we can easily calculate the absolute and relative errors of both devices (columns $AE_{E4}$ and $AE_{MsWH}$ for absolute errors and $RE_{E4}$ and $RE_{MsWH}$ for relative errors).

As can be seen both in Table 3 and Figure 18, the smallest error in the accuracy of the measurement taken with the Empatica wristband is as large as the biggest error made with the MsWH, which is better calibrated.

Once this point is verified, we proceed to perform a set of simultaneous measurements with both devices under equivalent measurement conditions and on the same individual. For this purpose, the alternative electrodes of the E4 will be used to measure EDA on the fingers. In a first experiment, the E4 is placed on the left hand and the MsWH is placed on the right hand, and then, the experiment is repeated by reversing the positions. By proceeding in this way, an attempt is made to minimize the effect of possible local differences in conductance between the two hands. Changes in skin conductance turn out to be very slow, so the duration of the experiment should be relatively long (just over 18 min).

The measurement processes of the two devices are different. As explained in Section 3.5.2, the MsWH uses an AC excitation signal at a frequency of 100 Hz. On the other hand, the E4 wristband uses a DC signal whose polarity is reversed between successive measurements. This is equivalent to saying that both devices measure skin conductance in different layers and may partly explain the experimental differences obtained. Similarly, the capacitance of the human body differs in measurements taken in DC and AC.

In Figure 19, the measurement results taken with the MsWH are shown. Being an AC signal, the results can be shown both in modulus (clear blue) and in phase (orange). This figure should be considered only as a sample of the raw results collected by the platform since, in its state, it cannot be compared with the results of the E4 wristband.

The MsWH module and phase data are combined to obtain a single reading, which is the real part of the signal obtained. This signal can be compared with that obtained with the E4 wristband. The results are shown in Figure 20.

As can be seen in Figure 20, although the numerical results differ notably (in the figure, the right and left axes differ in scale), their relative behavior (in terms of general trend and more or less abrupt variations) is quite similar.

## 6. Conclusions and Future Work

In the present work, a new multi-sensory platform for the collection of physiological signals, that will be used in emotion recognition, has been presented. The MsWH platform has been developed using both sensors and complete state-of-the-art measurement systems, seeking the highest precision in the measurement of physiological signals and the greatest versatility in the measurement and analysis processes available on the market. It is a device designed to record experimental data, so it is not intended as a commercial or wearable device. Although bulky, the complete platform is very light (168 g), and the battery has an autonomy of approximately 2 h with all the devices at full performance, which allows long-term experiments to be carried out, in which the amount of data recorded and analyzed are massive. Although the information can be partially analyzed within the system itself, the device has a high transfer rate Wi-Fi connection (including raw data as well as image and/or video), which allows the collected results to be sent to an external device for more detailed analysis.

The platform is divided into two subsystems. The first of them is fixed to the subject’s forehead with an elastic band. In it, the temperature of the skin, the EDA/bioimpedance and the inertial system (in addition to the camera) are recorded. The second subsystem is placed in the earlobe and measures blood oxygen saturation and heart rate. The situation of all of these sensors is optimal for several reasons. The forehead is a sufficiently smooth and flat surface to be able to fix the thermal sensor in almost permanent contact. In addition, it is one of the areas of the body where the EDA appears with greater amplitude, thus facilitating its measurement. In addition, the platform that supports the camera barely moves, so the facial reactions of the subject are accurately recorded. With regard to the earlobe, it is one of the most common areas for measuring oxygen saturation in the blood, after the fingers. Although many of the registered variables can be obtained in a wristband wearable system (more comfortable to wear and allowing the recording of, for example, arm movements, perhaps more efficient for expressing emotions), the truth is that it is usually more of a drawback than an advantage in the recording of other variables such as temperature or EDA, where vibrations and the movement of sensors and electrodes can be a source of significant noise. By placing these measure systems on the user’s forehead, the electrical noise associated with movement is reduced, without losing the ability to record movements and emotional reactions with our Attitude and Heading Reference System. Extending the use of this platform to the measurement of physical parameters with changes in intensity or physical activity will require appropriate sensor calibration and sensor validation.

Leaving aside the work of the porthole camera (whose vision area remains practically fixed, with the advantages that this entails), there are five signals recorded in total. With regard to skin temperature, a sensor has been used that requires thermal contact with the subject. Although the precision achieved is practically identical to that of Empatica, our measurement shows a high thermal inertia. The possibility of replacing the sensor presented here with another one based on contactless IR technology, with similar precision but faster response, has been assessed. Regarding the collection of EDA data, as has been said, the position of the sensors offers advantages with respect to, for example, the wrist (Empatica), since they are less sensitive to user movements. Multiplexing a dual analog signal increases the accuracy and ability to calibrate the device, and the use of an AC signal offers inherent advantages over DC or switched DC measurements. Furthermore, by placing the inertial measuring unit on the user’s head, the aim is to prioritize all kinds of gesticulations and movements that facilitate emotional classification in the future based on the set of physiological signals that this device is capable of capturing.

## Figures and Tables

**Figure 1 sensors-22-05775-f001:**
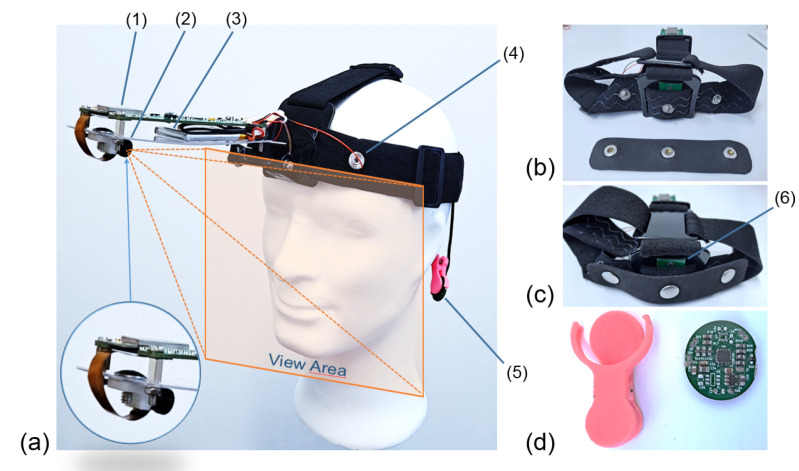
Final design of the platform and fastening system. (**a**) Side view of the platform. (**b**) Electrodes of the EDA/bioimpedance subsystem. (**c**) View with the electrodes placed in their final position and fixed by means of the EVA foam. (**d**) PCB with SpO2 and HR measure systems and fastening system in pink. (1) Microcontroller, (2) camera, (3) LIPO battery, (4) EDA electrode, (5) pulse oximeter, (6) temperature sensor.

**Figure 2 sensors-22-05775-f002:**
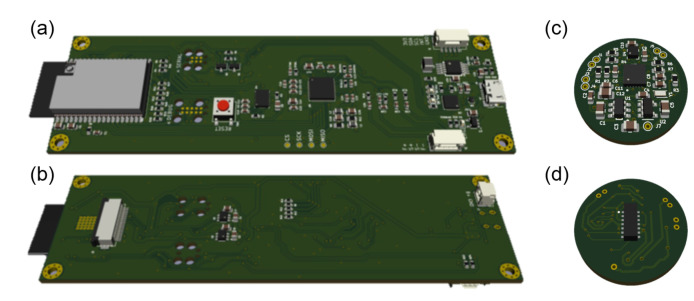
PCBs of the platform. (**a**) Top view of the main PCB, attached to the front of the device by means of a lightweight plastic bracket. It contains the power and communications control system (RS232 and Bluetooth, plus a USB port), the main microcontroller, the chip for EDA, the gyroscope and accelerometers (IMU) and the LIPO battery (including its charger) that powers all the electronics. (**b**) Bottom view of the same PCB, where the camera connection port and its power sources are located (left and center, respectively). The size of this PCB is 38 mm × 124 mm. (**c**) Top view of the pulse oximeter PCB, where the microcontroller and most of the components associated with the device can be seen. (**d**) Bottom view of the pulse oximeter PCB, where the corresponding microprocessor stands out. The size of this PCB is 18 mm × 20 mm.

**Figure 3 sensors-22-05775-f003:**
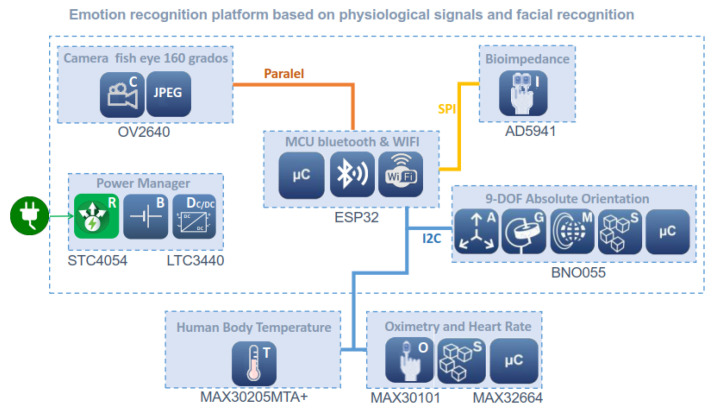
Block diagram of the complete sensoring and processing system.

**Figure 4 sensors-22-05775-f004:**
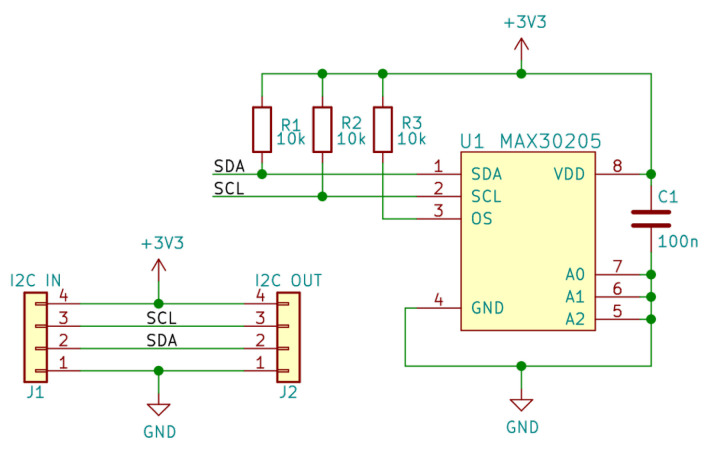
Circuit diagram for the ST sensor. This figure has been generated with KiCAD (https://www.kicad.org, accessed on 14 June 2022). The names in green are the names of the Integrated Circuits. In red are the pin numbers of the ICs.

**Figure 5 sensors-22-05775-f005:**
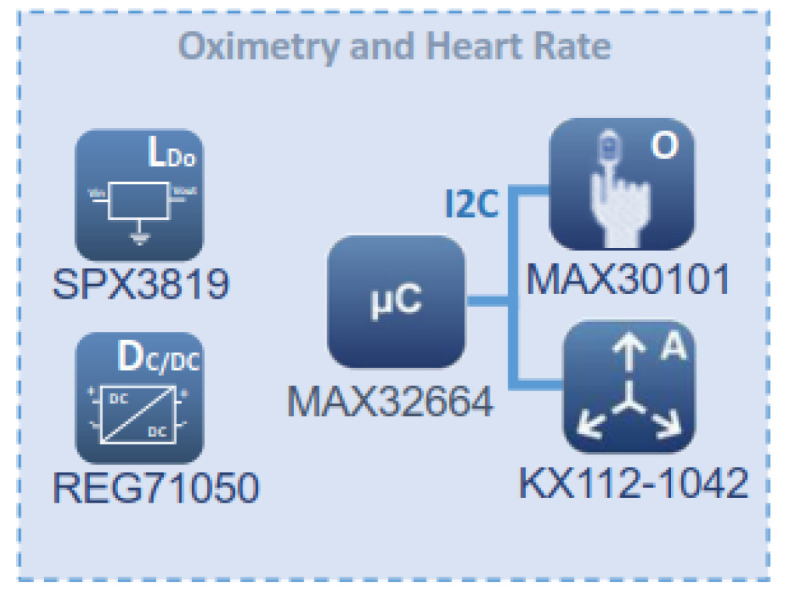
Block diagram of the blood oxygen saturation and heart rate monitoring subsystem.

**Figure 6 sensors-22-05775-f006:**
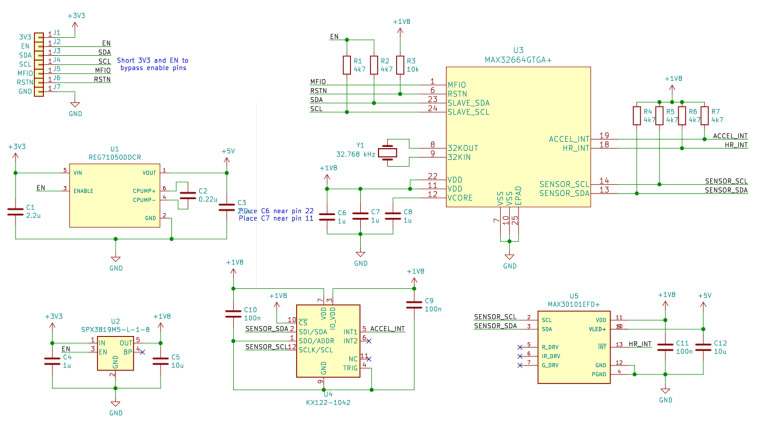
Circuit diagram for the SpO2 and HR monitoring subsystem. This figure has been generated with KiCAD (https://www.kicad.org, accessed on 14 June 2022). The names in green are the names of the Integrated Circuits. In red are the pin numbers of the ICs.

**Figure 7 sensors-22-05775-f007:**
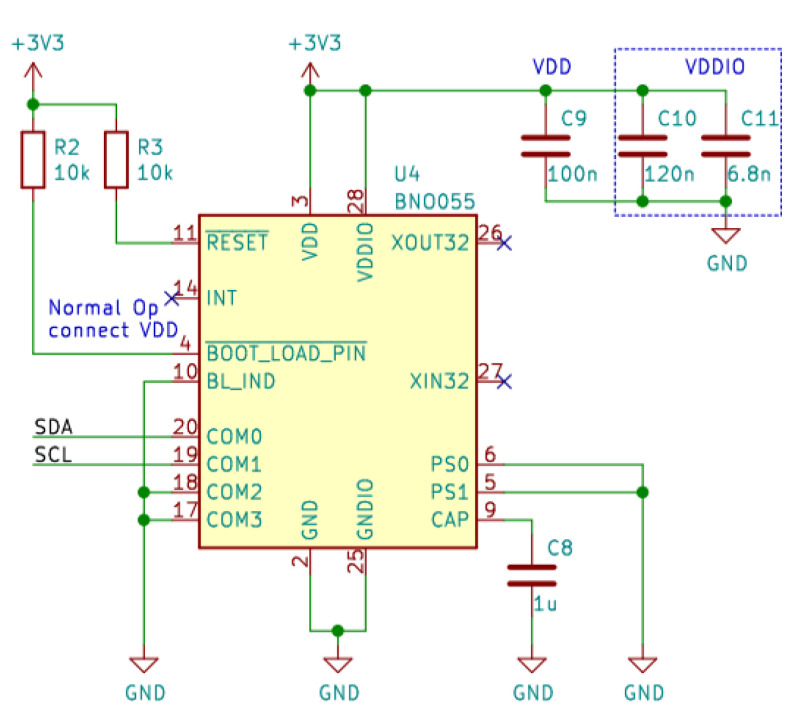
Circuit diagram for the IMU sensor. This figure has been generated with KiCAD (https://www.kicad.org, accessed on 14 June 2022). The names in green are the names of the Integrated Circuits. In red are the pin numbers of the ICs.

**Figure 8 sensors-22-05775-f008:**
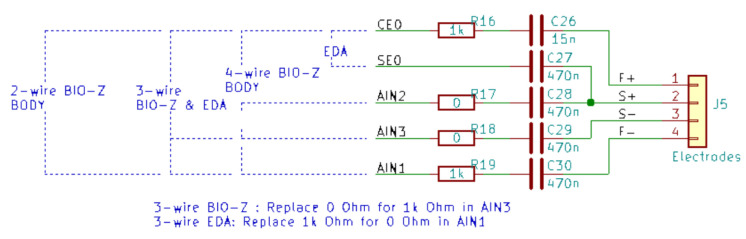
Connection of the electrodes for the three measurement modes, 2, 3 and 4 electrodes. Limiting/isolation resistors are included. This figure has been generated with KiCAD (https://www.kicad.org, accessed on 14 June 2022). See the text for an explanation of the two, three and four wire connection.

**Figure 9 sensors-22-05775-f009:**
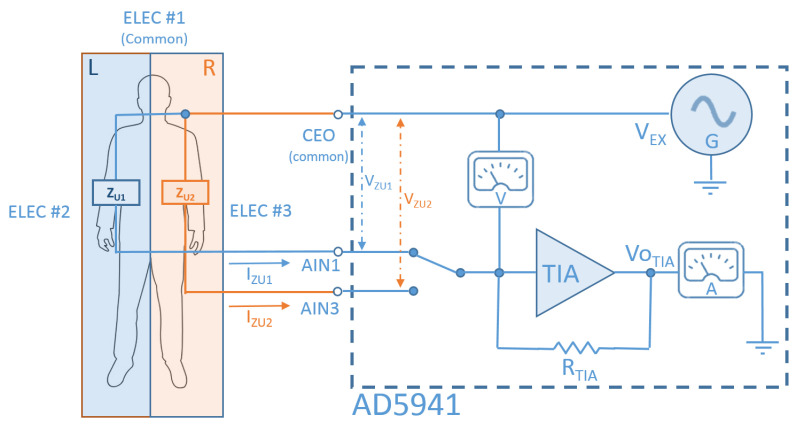
Block diagram for the measurement system. Observe the 3-electrode multiplexed configuration (see text).

**Figure 10 sensors-22-05775-f010:**
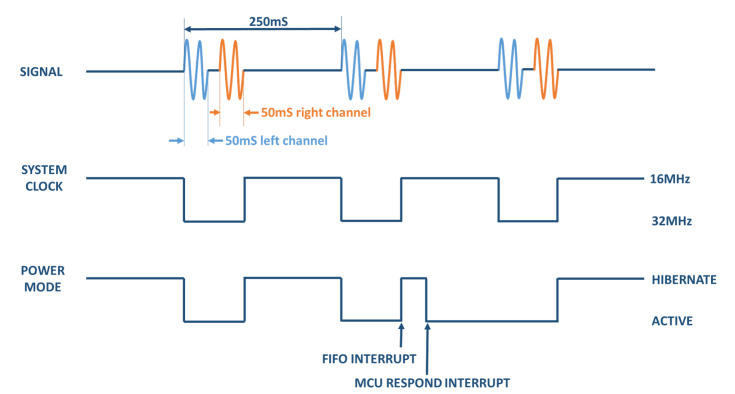
EDA measurement process. A complete cycle, including multiplexing and reading of the left (blue) and right (orange) channels, is completed in 250 ms–150 ms of which, approximately, the system remains hibernated. See text for further details. The interrupt signals correspond to the reaching of the FIFO memory threshold and the time at which the MCU receives the stored data.

**Figure 11 sensors-22-05775-f011:**
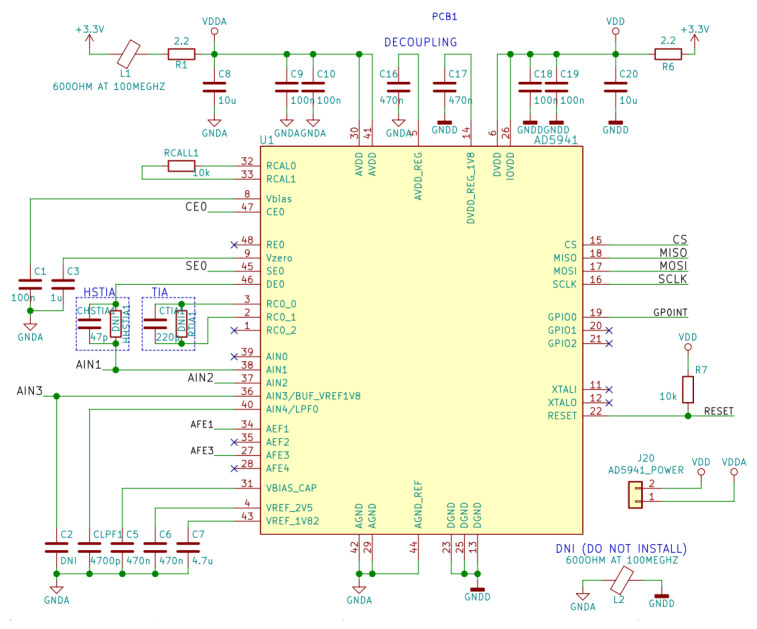
Circuit diagram for the EDA measurement unit, AD5941.

**Figure 12 sensors-22-05775-f012:**
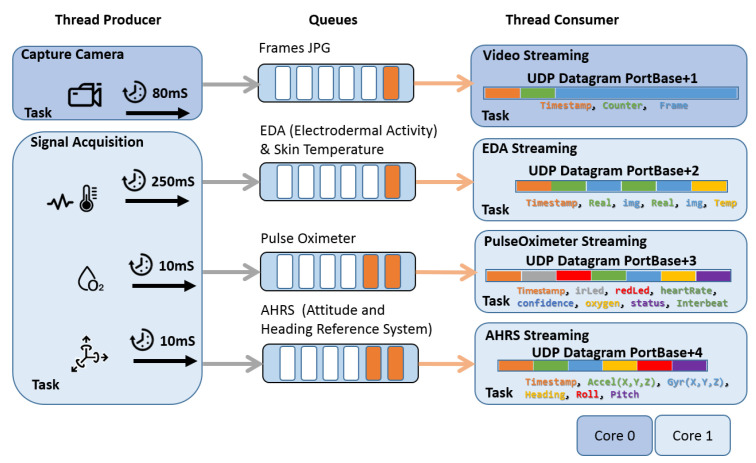
General description of the software and firmware of the developed platform.

**Figure 13 sensors-22-05775-f013:**
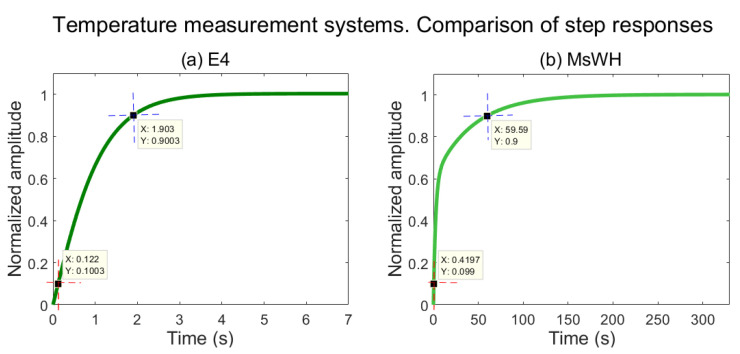
Step response of the body temperature measurement systems. (**a**) Empatica E4 wristband. (**b**) MsWH. $SRT_{E4}\approx 1.9$ s. $SRT_{MsWH}\approx 59.59$ s. Both subfigures have been obtained using the Matlab System Identification Toolbox.

**Figure 14 sensors-22-05775-f014:**
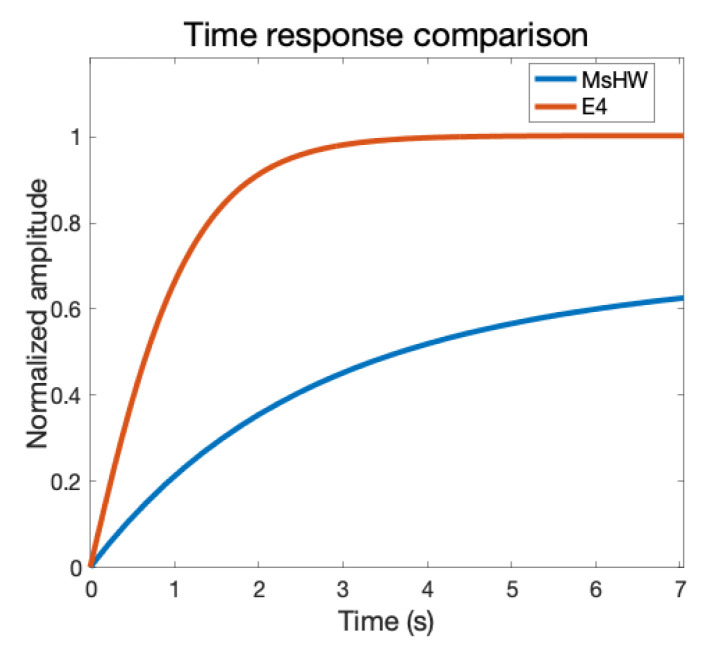
Time response comparison of the body temperature measurement systems. It can be clearly seen that the time response of the Empatica E4 wristband is much faster than in the case of the MsHW system. Horizontal axis: time in seconds. Vertical axis: Amplitude. Blue line: MsWH. Red line: Empatica E4 wristband.

**Figure 15 sensors-22-05775-f015:**
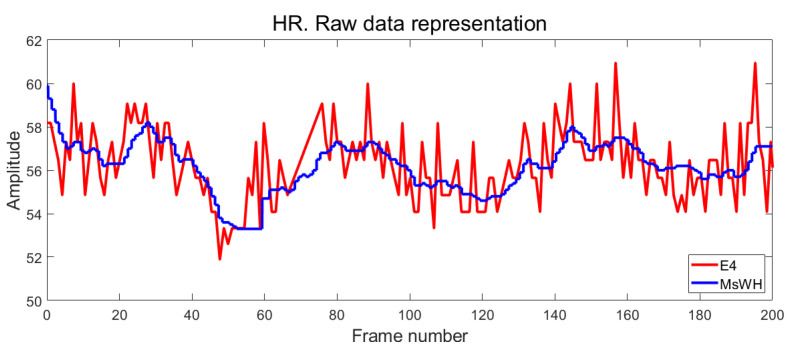
Representation of the raw data of the RH measurement. Horizontal axis: frame number. Vertical axis: Amplitude. Blue line: MsWH. Red line: Empatica E4 wristband.

**Figure 16 sensors-22-05775-f016:**
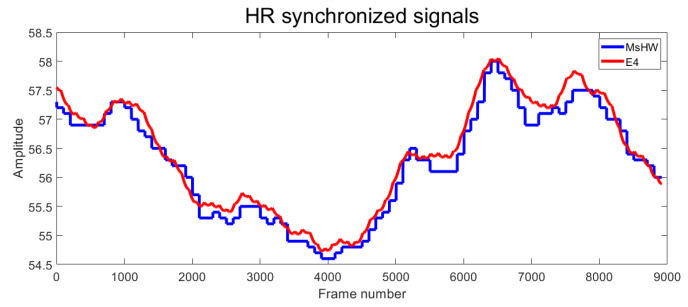
HR synchronized signals. Horizontal axis: frame number. Vertical axis: Amplitude. Blue line: MsWH. Red line: Empatica E4 wristband.

**Figure 17 sensors-22-05775-f017:**
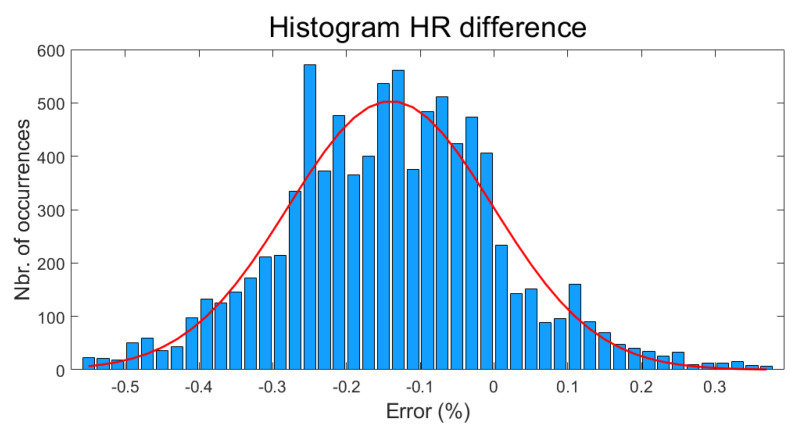
Difference between MsWH and Empatica E4 HR signals. Histogram. Horizontal axis: error (%). Vertical axis: Number of occurrences. Red line: Gaussian fit. The maximum is located at −0.14%.

**Figure 18 sensors-22-05775-f018:**
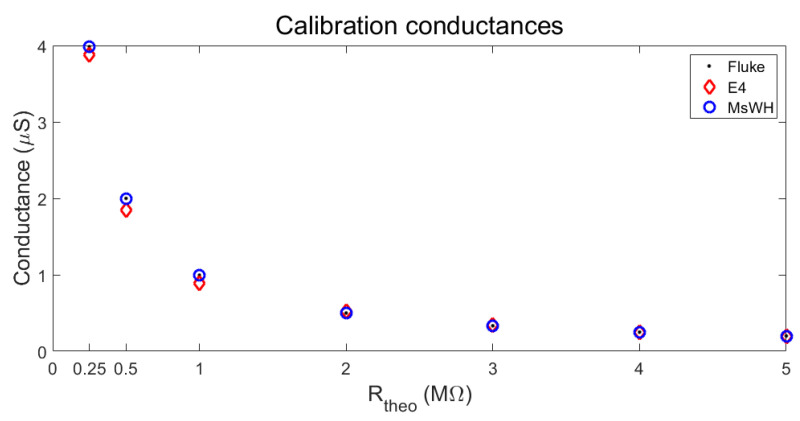
Conductance calibration. Experimental results. MsWH and E4.

**Figure 19 sensors-22-05775-f019:**
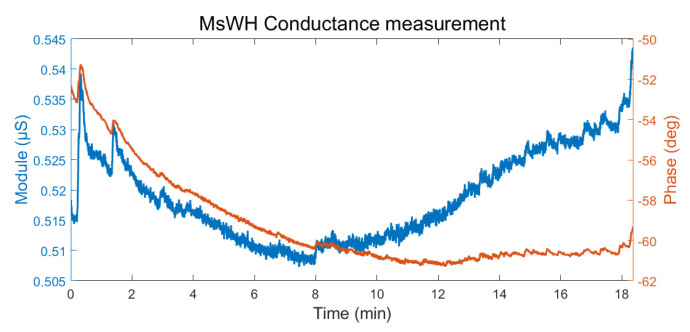
MsWH conductance measurement. Raw data. Left axis: module (μS). Right axis: phase (degrees).

**Figure 20 sensors-22-05775-f020:**
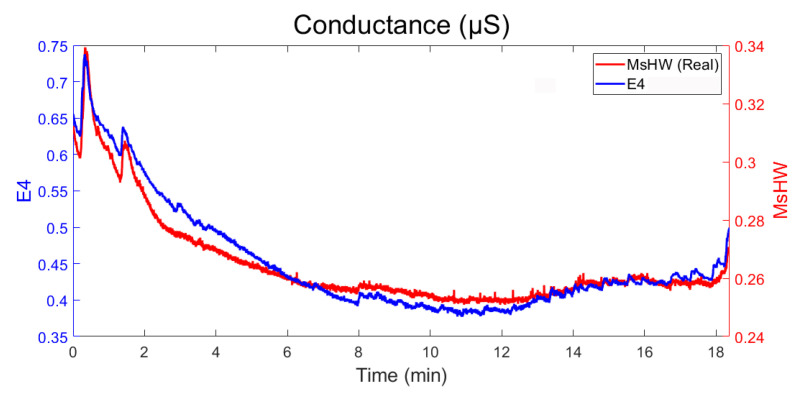
Comparison of measurement of skin response conductance using MsWH and E4 over the same subject. Blue line (left axis): E4 measurement (μS). Red line (right axis): MsWH measurement (μS).

**Table 1 sensors-22-05775-t001:** Sensors and main characteristics. N.L.→Nonlinearity. S.R.→Sample rate. Acc.→Accuracy. The sample rate of the AD5941 is 800 ksps. Data are filtered and downsampled in order to obtain a final data sequence at only 4 Hz, which is enough to register EDA activity.

Sensor	Device/Measure	Range	Resolution (#bits)	Bandwidth (Hz)	Other
	**Accelerometer**	±2 g/±4 g ±8 g/±16 g	14	8, 16, 31, 63 125, 250, 500, 1000	N.L.: 0.5%FS
**BNO055**	**Gyroscope**	±125, ±250, ±500 ±1000, ±2000 (°/s)	16	12, 23, 32, 64 47, 116, 230, 523	N.L.: ±0.05%FS
	**Magnetometer**	±1300 μT X, Y axis ±2500 μT Z axis	13/13/15	2, 6, 8, 10 15, 20, 25, 30	N.L.: 1%
**MAX30205**	**ST**	0 °C to +50 °C	16	20 (max.)	Acc.: 0.1 °C (37–39 °C)
**MAX30101**	**SpO2 + HR**	640–980 nm	18	50–3200	-
**AD5941**	**EDA**	100 kΩ–10 MΩ 10–0.1 μS	16	4	S.R.: 800 ksps, Acc.: 1%

**Table 2 sensors-22-05775-t002:** Sensors and main power consumption characteristics.

Device/Sensor	Theoretical	Measured	Sample Rate
		(mA)	(fps)
**MAX30205 (ST)**	600 μA	0.604	4
**MAX30101 + MAX32664 (SpO2 + HR)**	-	9.8 to 19.5	100
**AD5941 (EDA)**	-	1.4	4
**ESP32-WROVER-IE**	500 mA	100	-
**OV2640**	42.7mA	53.2	12.5
**BNO055**	12.3 mA	12.3	20

**Table 3 sensors-22-05775-t003:** EDA measurement. Conductance calibration. Experimental results. First column: Theoretical value of the resistance set. Second: related conductance values. Third: high-precision measurement of resistances. Fourth: related high-precision conductance values. Fifth and sixth: experimental conductance measurements through the E4 wristband and MsWH, respectively. Seventh and nineth: absolute and relative errors for the E4 wristband. Eighth and tenth: absolute and relative errors for the MsWH.

$R_{\mathrm{theo}}$ (Ω)	$S_{\mathrm{theo}}$ (μS)	$R_{F}$ (Ω)	$S_{F}$ (μS)	$S_{E4}$ (μS)	$S_{\mathrm{MsWH}}$ (μS)	${\mathrm{AE}}_{E4}$ (μS)	${\mathrm{AE}}_{\mathrm{MsWH}}$ (μS)	${\mathrm{RE}}_{E4}$ (%)	${\mathrm{RE}}_{\mathrm{MsWH}}$ (%)
250,000	4.0	251,000	3.984064	3.88	3.987	0.1042	−0.0031	2.615	−0.079
500,000	2.0	500,001	1.999996	1.85	1.999	0.1484	0.00072	7.4188	0.036
1,000,000	1.0	1,006,000	0.994036	0.889	0.9968	0.1047	−0.0028	10.531	−0.276
2,000,000	0.5	2,000,500	0.499875	0.5188	0.499	−0.01897	0.00089	−3.795	0.178
3,000,000	0.33	3,001,100	0.333211	0.342	0.332	−0.0088	0.0012	−2.654	0.353
4,000,000	0.25	4,001,000	0.249938	0.2523	0.249	−0.0024	0.00096	−0.956	0.386
5,000,000	0.2	5,001,200	0.199952	0.1985	0.1988	0.0014	0.0011	0.689	0.563

## Data Availability

Not applicable.

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
