# Peer review of "MsWH: A Multi-Sensory Hardware Platform for Capturing and Analyzing Physiological Emotional Signals"

_sensors, 2022, doi:10.3390/s22155775_

Round 1

Reviewer 1 Report

  • This paper presents a new physiological signal acquisition multi-sensory platform for emotion detection. The system can record and analyze five physiological signals: skin temperature, blood oxygen saturation, heart rate (and its variation), head movement/position of the user, and electrodermal activity/bioimpedance. A porthole camera complements the measurement system for facial expression recognition. 
  • The paper describes the circuitry of all different components and how they are interconnected. There is also a brief description of the software developed for running the system. The system is interesting, but some issues need to be addressed.
  • First of all, the system presented has to be validated with a series of well-defined tests conducted with accurate measurement systems on a test bench and not just comparing it with a similar device. 
  • The SpO2, the head IMU and the camera were not tested.
  • The camera mounts a fisheye that most probably distorts the image, so it has to be proved for image recognition for facial expression recognition.
  • The authors report contradicting information in the paper for the measurement of bioimpedance. In some places (lines 216 and 276), it is reported that they use a square wave at 100Hz, while on lines 265 and 280, they say that it is a sinusoidal wave of 1.1 amplitude at 100Hz. So, which is the right one? If it is a square wave, which are the amplitude, the duty cycle, and the type (monophonic, biphasic). In the case of a sinusoid, how do they generate it?
  • Again for computing the bioimpedance, why do they use a DFT for computing the imaginary part and not just use the phase shift between voltage and currents? How do they compute the DFT on the device? Moreover, why do they use a 3-wires configuration? The two different impedance computed from ref. And AN3 and from ref. And AN1 have different value sine the two distance are different.
  • Figure 18 shows the measurement results of dermal impedance taken with the proposed system on a human being. It shows the results both in modulus and in phase. However, since there is no term of comparison with a deterministic measuring system, it has no validity at all.
  • Figure 19 shows the comparison of the measurement of skin response impedance using the proposed system and the Empatia E4 over the same subject. They differ considerably in amplitude. So, again, without comparison with a deterministic measuring system, it has no validity at all. Moreover, the amplitude of the signal given by the proposed system is almost double the one provided by the E4. 
  • About the software. Why do they use the UDP protocol for all data transmission? UDP is not reliable for data transmission since it is a connectionless protocol.
  • Table1 reports the sensor’s main characteristics. I doubt that the bandwidth is expressed in kHz.
  • Table2 reports the sensors’ main power and consumption characteristics. Pay attention that are not only mA but also mW (line relative to the OV2640)
  • In line 31, it is not arterial oxygen saturation but only oxygen saturation.
  • In line 235, there are two times: “in the”
  • In line 287 at the beginning, it is 150ms and not 150m.
  • In line 334, it is a 3-axis accelerometer, not 3 accelerometers.

Reviewer 2 Report

A new physiological signal acquisition multisensory platform for emotion detection is reported in this manuscript.  It will be of interest to the readers of Sensors.  The authors are encouraged to address this issues listed below.

It should be clarified that although this is a “wearable system”, it is appropriate for clinical experiments in laboratories not for daily life in free living.

Please make a table to define all abbreviations used

The difference in the scale of time gives the impression that MsWH is responding faster than E4. A third subfigure with the both responses on the same timescale should be added for a better visual comparison.

Are the same signal processing algorithms used for E4 and MsHW?

Would it be possible to compare these HR with higher accuracy systems such as BioPlux or with EKG to provide a “ground truth”?  In our experience the HR reported by E4 is based on different algorithms depending on reporting in real-time streaming data or processing batch data. 

HR accuracy changes with changes in intensity of physical activity.  One can assume the same behavior depending on the emotional inducement.  It would be useful to check HR accuracy for various levels of intensity (if emotion inducement is difficult, by intensity of physical activity)

Page 18: Assessment of benefits of using head movement vs. arm movement?  A wristband would not interfere with movements of arms or fingers.

The potential of the proposed hardware is beyond use for capturing emotional signals.  It would be useful for detecting and discriminating acute psychological stress and physical activities occurring at different times or concurrently.  Some publications that illustrate such needs:

·      Sevil M, M Rashid, I Hajizadeh, M Park, L Quinn, A Cinar. Physical Activity and Psychological Stress Detection and Assessment of Their Effects on Glucose Concentration Predictions in Diabetes Management, IEEE Trans on Biomedical Engineering, 2021, 68(7):2251-2260.  DOI:10.1109/TBME.2020.3049109. PubMed ID: 33400644

Reviewer 3 Report

The paper proposes a platform for the acquisition of physiological signals, focusing on both the hardware used (sensors, conditioning, microprocessors, connections) and the software, optimized for accurate and massive data acquisition.

In my opinion, the paper needs to be improved.

First of all, it is necessary to improve the discussion on works that address the development of platforms for physiological signal acquisition. A brief discussion is presented in the Introduction section. The paper can be improved by adding a section on related wors. In this section, the works cited in the introduction section need to be better discussed. In particular, the authors should provide a discussion characterizing each work, its benefits and drawbacks, and a comparison among the related works to justify the necessity of the platform proposed in the paper. Please state clearly and precisely in the paper what makes this work original.
